# Su(H) Modulates Enhancer Transcriptional Bursting in Prelude to Gastrulation

**DOI:** 10.3390/cells13211759

**Published:** 2024-10-24

**Authors:** Kelli D. Fenelon, Priyanshi Borad, Biraaj Rout, Parisa Boodaghi Malidarreh, Mohammad Sadegh Nasr, Jacob M. Luber, Theodora Koromila

**Affiliations:** 1Department of Biology, University of Texas at Arlington, Arlington, TX 76010, USA; kelli.fenelon@uta.edu (K.D.F.); phb9243@mavs.uta.edu (P.B.); 2Department of Computer Science, University of Texas at Arlington, Arlington, TX 76010, USA; bxr1886@mavs.uta.edu (B.R.); pxb7330@mavs.uta.edu (P.B.M.); jacob.luber@uta.edu (J.M.L.); 3Multi-Interprofessional Center for Health Informatics, University of Texas at Arlington, Arlington, TX 76010, USA; 4School of Biology, Aristotle University of Thessaloniki, 54124 Thessaloniki, Greece

**Keywords:** transcriptional bursting, gene expression dynamics, Su(H), Notch/Hairless signaling, sog enhancer, embryogenesis

## Abstract

Transcriptional regulation, orchestrated by the interplay between transcription factors (TFs) and enhancers, governs gene expression dynamics crucial for cellular processes. While gross qualitative fluctuations in transcription factor-dependent gene expression patterning have a long history of characterization, the roles of these factors in the nuclei retaining expression in the presence or absence of these factors are now observable using modern techniques. Our study investigates the impact of Suppressor of Hairless (Su(H)), a broadly expressed transcription factor, on enhancer-driven transcriptional modulation using *Drosophila* early embryos as a model system. Building upon previous findings, we employ super-resolution microscopy to dissect Su(H)’s influence on *sog-Distal* (*sogD*) enhancer activity specifically in nuclei with preserved *sogD*-driven expression in the absence of Su(H) binding. We demonstrate that Su(H) occupancy perturbations alter expression levels and bursting dynamics. Notably, Su(H) absence during embryonic development exhibits region-specific effects, inhibiting expression dorsally and stabilizing expression ventrally, implying a nuanced role in enhancer regulation. Our findings shed light on the intricate mechanisms that govern transcriptional dynamics and suggest a critical patterning role for Notch/Hairless signaling in *sog* expression as embryos transition to gastrulation.

## 1. Introduction

In eukaryotic systems, gene transcription orchestrates crucial cellular processes through intricate interactions between proteins and DNA, often leading to intermittent bursts of RNA production [1,2]. This complex process commences with specific transcription factors (TFs) binding to DNA motifs situated within regulatory elements, such as promoters or enhancers. Despite advancements in understanding enhancer-mediated gene regulation, the precise contribution of TFs at enhancer sites to incongruities between controlling factor and target gene patterning remains a topic of ongoing investigation [3,4,5,6,7]. The question of whether TFs engage in competitive interactions with activators to establish a binary transcriptional state or exert regulatory control over transcriptional rates through the modulation of enhancer–promoter contact (EPC) or RNA polymerase recruitment remains murky.

Moreover, recent studies have popularized the investigation of transcriptional bursting in *Drosophila* embryos, shedding light on gene expression dynamics during development [8,9,10,11]. This phenomenon, characterized by intermittent transitions between periods of heightened transcriptional activity and dormancy, represents a fundamental aspect of gene regulation [5,12,13]. Studies have revealed that transcriptional bursting is prevalent during *Drosophila* embryonic development and genes involved in developmental processes often exhibit burst-like patterns of transcription, leading to dynamic changes in mRNA levels over time. These processes are regulated by a combination of transcription factors, chromatin modifiers, and enhancer elements [14,15].

Utilizing traditional and state-of-the-art methodologies, including single-molecule fluorescent microscopy techniques like the MS2 [16,17] and PP7 systems, which capitalize on coat [18,19,20,21] proteins coupled with fluorophores, researchers have elucidated many subtleties of transcriptional bursting during *Drosophila* embryogenesis [22,23]. These techniques enable the visualization and quantification of transcriptional dynamics at the single-cell level in real time [24,25,26,27,28].

Enhancers, pivotal for orchestrating gene expression patterns and spatial outcomes, exemplify dynamic regulatory entities [29]. For example, the *sogDistal* (*sogD*) enhancer plays a crucial role in regulating the expression of *short gastrulation* (*sog*) during *Drosophila* embryonic development. This regulatory paradigm extends beyond *sogD,* which showcases alterations in transcription factor binding dynamics, notably exemplified by Suppressor of Hairless (Su(H)), which assumes a repressive role on the dorsal side of the embryo during cellularization [28,30].

Despite its repressive role being limited to the dorsal side of the embryo, *Su(H)* is ubiquitously expressed until cellularization, when a slight anteroposterior pattern begins to emerge [28]. However, it is well established that Su(H) facilitates either repression or activation through interaction with cofactors Hairless [31] (H) and Groucho [32] (Gro) or Notch [33,34] (N) and Mastermind [35] (Mam), respectively, and that Notch signaling is necessary for the expression of ventral genes, e.g., snail [36]. Further, Notch has been shown previously to affect transcriptional bursting dynamics in other contexts [37,38]. In our present study, we expand upon existing knowledge by elucidating the dynamics inherent in enhancer-driven patterning in *Drosophila* embryos. Specifically, we seek to unravel the nuanced role of Su(H) in shaping enhancer dynamics and developmental gene regulation mechanisms within spatiotemporal regions in which target gene expression persists in its absence and its effect on sporadic bursts of RNA transcripts [28]. Leveraging single-molecule quantitative live imaging assays, employing Zeiss LSM 900 Airyscan 2 super-resolution microscopy, we explore the potential involvement of Su(H) in transcriptional bursting phenomena and its role in the precise spatiotemporal regulation of gene expression during *Drosophila* embryogenesis.

## 2. Materials and Methods

### 2.1. Fly Husbandry and Embryo Preparation

Virgin females maternally expressing MCP-GFP (green) and Nup-RFP (red) were crossed with either *sog_Distal eve2 promoter-MS2.yellow-attB* [30] or *sogD_ΔSu(H) eve2 promoter-MS2.yellow-attB* [28] males. Embryos were timed and individually collected during nc10–11. Collected embryos were carefully dechorionated and mounted in Halocarbon 27 between the slide and coverslip spaced with double-sided tape, as previously described [28].

### 2.2. Live Imaging

Embryos were collected in apple agar plates for 1 h and then they were rested at RT for 30 min before imaging. Prior to this, all embryo collection bottles were subjected to a 30 min pre-lay period to synchronize the collection process by clearing the female flies. The embryos were collected by hand-dechorionation, and then placed between a glass slide and a coverslip with a heptane-dissolved adhesive. The assembled embryos were subsequently immersed in Halocarbon 27 oil [28]. Embryos were imaged using super-resolution microscopy during the stages leading into gastrulation on a Zeiss LSM 900 Airyscan 2 (Zeiss, Oberkochen, Germany). Broad-view movies were captured using the 40× water oil immersion objective; we took super-resolution movies with 1× zoom, a 3% 488 nm laser, and a 2.2% 555 nm laser. Frames were captured in 4096 × 4096 px resolution in stacks of 25 (0.9 µm interval) every 3.98 s. Super-resolution movies were captured using a 40× water oil immersion objective, 4.5× zoom, a 3% 488 nm laser power, and a 2.2% 555 nm laser power. Frames were captured at a 700 × 700 px resolution in stacks of 7 (0.5 µm interval) every 1.77 s, centered at approximately the midpoint (50%) of the embryo’s length from a lateral view.

### 2.3. Tissue Image and Statistical Analyses

Projected confocal stack series of a live Drosophila embryo were flattened and color-balanced, as previously described [28], and then bleach-corrected (https://github.com/fiji/CorrectBleach, accessed on 9 October 2024) in ImageJ (v1.54). The resulting 2D movies were then divided into separate stages by nuclear cycle (identified by counting backward from nc14) to include the very last frame containing a visible connection between nuclei. These movies were then processed using various image processing techniques in OpenCV [28] to track MS2 dot maturation and movements using in-house software. First, the movie was split into frames in jpeg format. For all data, we applied a threshold of 55 for the green channel to remove noises. Then, the mean pixel value for each contour was calculated and normalized with minimum and maximum average pixel value of contours across a certain stage.

Deep statistical analysis for the measurement of statistical data such as mean, median, and standard deviation was performed for each stage in the case as well as for control cohorts, with metrics such as the number of green regions, mean size of green regions, total size of green regions, mean brightness, and mean normalized brightness. In order to compare case and control, two different statistical testing scenarios were used: Mann–Whitney U Test and permutation test. In the Mann–Whitney U Test [39], the primary interest is in whether one set tends to have a statistical metric larger or smaller than the other set, so we fed case and control datasets and found *p*-values that were significant. The permutation test [40] does not assume a normal distribution and is suitable for comparing distributions of statistical metrics between the two sets. The null hypothesis (H0) is that the distribution of statistical metrics is the same across the two sets and the alternative hypothesis (H1) is that the distribution of statistical metrics is greater for one than the other, which can be confirmed using the mean observed difference in case and control data for each metric. Simply put, rejecting the null hypothesis means that we have enough statistical evidence to accept the observed difference between case and control. If the observed difference is positive, we infer that the case group is greater than the control group—otherwise, control is greater, assuming the *p*-value from the permutation test is less than 0.05.

All other *p*-values were calculated using unpaired *t*-tests unless variance between conditions were significant, in which case Welch’s *t*-test was used. Error bars are SEM, unless otherwise stated.

### 2.4. Nuclei Tracking and Quantification

Z-projected confocal stack series of a live *Drosophila* embryo were flattened using maximum intensityand color-balanced, as previously described [28], and bleach-corrected (https://github.com/fiji/CorrectBleach, accessed on 9 October 2024) in ImageJ. The resulting 2D movies were then divided into separate stages by nuclear cycle (identified by counting backward from nc14) to include the very first frame containing visually detectable anaphase initiation. The tracking algorithm used here leverages both spatial and temporal information to maintain a continuous identification of objects across frames. This algorithm works by identifying green dots inside nuclei in each frame, calculating their centroid, and then comparing the distance of each centroid to those of dots from the previous frame with a defined distance threshold. In fact, we assigned unique IDs based on proximity to dots in previous frames. This approach allowed us to track the movement of nuclei and brightness changes in the green dots across successive frames accurately.

## 3. Results

Broadly expressed transcription factors involved in repression of the *sogD* enhancer of the *sog* gene, such as Su(H), form anteroposterior (AP) patterns of expression [30]. However, in stark contrast to the eventual ubiquitous AP expression pattern of Su(H), Su(H) regulates the *sogD* expression pattern by limiting the transcription of *sog* in the dorsal fraction of the embryo and *sogD*-driven expression expands dorsally when Su(H) binding motifs in the enhancer region are mutated (Figure 1A,B, Appendix A) [28,30]. It is well known that the *sogD* enhancer is co-occupied by several transcription factors to produce the wildtype *sog* expression pattern through combinatorial action (Appendix A (see Appendix A)). We sought to interrogate the effect and mechanism of perturbations to Su(H) binding at enhancer regions by quantitatively analyzing deviations from wildtype embryonic expression dynamics of *sogD* in the absence of Su(H) occupancy in fine temporal resolution using super-resolution microscopy.

In 2019, Koromila and Stathopoulos showcased that Su(H) governs the spatial expression of *sog* rather than its transcription initiation, which is overseen by a different broadly expressed repressor, Runt. However, if perturbing Su(H) occupancy modulates EPC efficiency, we would expect to see fluctuations in the timing of transcriptional start following each nuclear cycle (nc) division in the absence of Su(H) enhancer occupancy (Figure 1C,D). To test this, we sought to image spatiotemporal MS2-MCP:GFP dynamics specifically in regions where *sogD* drives MS2 expression under both wildtype (wt) and Su(H) mutant conditions [28], targeting a resolution that balanced the need to visualize many individual nuclei in super resolution as the MZT concludes and cellularization and gastrulation begin (nc12–14, Figure 1E–G). Thus, we narrowed the imaging area to a 40 μm × 40 μm window positioned directly anterior to the center of the *sog* expression pattern as to overlap the wt and mutant expression domains (see Figure 1G). Indeed, we observed an apparent shift in post-division distributions in transcription initiation in reporter constructs lacking Su(H) binding sites at nc12 (Figure 2A, Appendix A).

On the other hand, if Su(H) occupancy perturbations ultimately modulate RNA polymerase recruitment, we would expect to see altered maturation of fluorescent reporter puncta as polymerase molecules are recruited at the promoter in the absence of the factor. In agreement with this, fluorescent dot initiation seems to hasten in the absence of Su(H) during nc12, but seems to dissipate or reverse as the MZT matures (Figure 2A). Interestingly, maximal fluorescent dot maturation time in the absence of Su(H) binding appears to be decreased during the next nuclear cycle at nc13 (Figure 2B). These apparent complementary changes prompted us to ask whether Su(H) absence is acting to destabilize EPC stochastically within our imaging region. As expected, however, we were unable to detect an obvious change in range of the total number nor in the time it takes for activation of all nuclei in the absence of Su(H) due to the saturation of active nuclei within our wt/mut window (Figure 2C).

Consistent with these observations, deep statistical analysis (Figure 2D–I) shows that the average rate of transcription during nc12 is significantly increased in *sogD^ΔSu(H)^* mutant reporters, as measured by dot size, which reflects nascent transcript number (Figure 2D). Further, consistent with Su(H) acting to delay post-nuclear division transcriptional activity, during that early stage, the initial burst in active nuclei occurs significantly earlier in mutant reporter embryos (Figure 2E). Subsequently, as zygotic gene expression increases in the next nuclear cycle, nc13, this early start is replaced by a significantly accelerated initiation of all nuclei in the absence of Su(H) binding (Figure 2F). Furthermore, on the trailing edge of the maternal to zygotic transition (MZT), during nc14, the effect of Su(H) exclusion emerges as a decrease in initial nuclei activation and slower initial transcription within the nuclei (Figure 2G–I) within the expression imaging window. Finally, we observed that Su(H) likely plays a role as a destabilizing factor as it significantly influences transcription rate variance both early and late and transcriptional activation time variance during nc13 (Appendix A).

As it is likely for subtle effects to be masked by stochastic transcriptional dynamics, we next sought to investigate the effect of Su(H) binding on the transcriptional dynamics of the individual nuclei in this spatiotemporal region. Accordingly, the spatiotemporal resolution of our methods facilitated the robust tracking of individual nuclei through each nuclear cycle through consistent and precise nascent transcript detection (Figure 2J–L, Appendix A). Stochastic transcriptional dynamics were immediately obvious (Figure 3).

These stochastic dynamics were especially conspicuous in transcriptional bursting events (Figure 3B,E,H–J). The perturbation of transcriptional factors at enhancers has been shown to modulate transcriptional bursting dynamics [1,10,41]. Interestingly, comparison of our *sogD* and *sogD^ΔSu(H)^* datasets revealed apparent fluctuations in tissue-level transcript density in both reporters at all stages (Appendix A). This implies that these seemingly random distributions of individual bursting events may coalesce to affect tissue-level transcript fluctuations, but future studies are required to resolve these spatiotemporal hotspots.

On an individual nucleus level, the effect of Su(H) binding increases as the MZT progresses, having little or no effect during nc12, but affecting all phenomena measured during nc14 (Figure 4A). Contrary to the effect of Su(H) binding in the dorsal lateral embryo where Su(H) absence at *sogD* induces transcription [28], we found that fluorescent dots become significantly smaller and less bright in *sogD^ΔSu(H)^* than in *sogD* embryos as the MZT approaches conclusion (Figure 4B–D). Further, dots at nc14 start later and persist through fewer frames in the absence of Su(H) binding, implying a region-specific activation role for Su(H) in the ventral half of the lateral embryo (Figure 4E,F). Concurrently, nuclei of *sogD^ΔSu(H)^* embryos burst more often for shorter durations, implying further transcriptional reduction through EPC destabilization (Figure 4G–I). This destabilization begins as early as nc13 and persists into gastrulation as dot brightness and size distributions are significantly altered (Figure 4J,K). Together, these data paint a clear picture showing that Su(H) functions within the *sogD* enhancer to stabilize and increase *sog* expression within the wildtype *sog* expression domain, demonstrating a dual role for Su(H) in *sog* regulation inside versus the tissue dorsal of the wildtype expression domain.

## 4. Discussion

The *sogD*-driven expression pattern is expanded dorsally just prior to gastrulation when Su(H) binding sites are mutated. The regulatory dynamics of Su(H) within the *sogD* domain are difficult to assess in aggregate owing to stochastic distributions of an attenuating, and not a precluding, mutant effect. However, we detect, ~30 min after *sog* expression initiation (nc13+), reduced transcription in the absence of Su(H) binding within the population of individually tracked nuclei existing in the overlap of wt and *ΔSu(H)* expression domains. At nc14, leading into gastrulation, transcripts are fewer, transcription begins later, and transcription bursts are more frequent and shorter lived in the absence of Su(H). This strongly implies Su(H) acts to bolster robust transcription in these nuclei, contrary to its role in transcriptional repression dorsally.

Interestingly, earlier in development, when the expression pattern is less affected by Su(H) perturbations [28], this pro-expression role of Su(H) is seemingly attenuated or absent. Our findings point towards a potential stage-dependent role for Su(H) in enhancer regulation. Such stage-dependent roles for early TFs have been shown previously [28]. For example, Runt, a well-known repressor, has been shown to act effectively as an activator in some contexts [28], while Zelda, a well-known activator, has been shown to have an effectively repressive role in specific instances [42]. In both cases, these factors were shown to function outside their well-established roles during major shifts in factor availability near or during regulatory hand-offs. Pre-gastrulation differentiation is an understudied phenomenon [43,44] and our findings here underscore the importance of considering temporal dynamics and stage-specific effects when studying enhancer–promoter interactions and transcriptional regulation.

These novel findings are not altogether surprising as Notch signaling is known to be active in the ventral embryo at nc14 [36,37]. Together, our data support a model whereby the shifting epigenetic landscape and morphogen populations during the MZT engenders the shifting of ultimate roles for the Su(H) control of *sog* expression as embryonic development progresses (Figure 5). In this model, early Su(H) action in this population is mild or antinomious in the early nuclear cycles(Figure 5A–F). Next, as maternally loaded factors diminish and zygotic expression begins, Su(H) begins to have a more measurable effect in these nuclei, likely resulting from the initiation of Notch signaling at nc13 (see Figure 5G–J). Finally, as the MZT handoff nears completion, Su(H) acts to facilitate transcriptional activation in this region, producing higher rates in its absence (Figure 5K–M).

In summary, our investigation sheds light on the molecular mechanisms governing the regulation of *sog* expression by Su(H), revealing the spatiotemporal nuance of function. Excitingly, the advanced, modern imaging techniques employed here promise to demystify the subtle roles of Su(H) in additional contexts as well as other factors through clever innovation in experimental and computational techniques [45,46]. Future research endeavors focusing on elucidating the precise molecular pathways through which Su(H) influences enhancer–promoter contacts and transcriptional bursting dynamics at these early stages will further enhance our understanding of gene regulatory networks underlying embryonic patterning and development.

## Figures and Tables

**Figure 1 cells-13-01759-f001:**
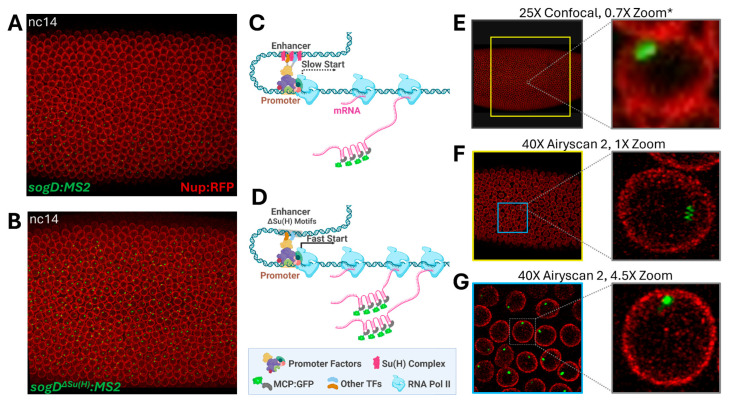
Imaging of *sogDistal* (*sogD*) enhancer dynamics. (**A**) A snapshot from confocal movie of sogD:MS2 at nc14 showing expression in green dots (MCP.GFP). (**B**) A snapshot from a confocal movie of *sogD^ΔSu(H)^*:MS2 at nc14 showing expression in green dots. (**C**,**D**) A schematic diagram of the simplistic repressor regulation model. When the enhancer is repressed, E-P contact is suppressed, resulting in delayed or reduced transcription (**C**). In the absence of repressor occupancy, E-P contact is more robust resulting in efficient transcription (**D**). (**E**–**G**) Narrowing the window of image acquisition allows for optimal single-nucleus resolution while capturing many cells. The yellow box in (**E**) represents the imaging window area at 40×, 1× zoom, as depicted in (**F**). The blue box in (**F**) represents the imaging window area at 40×, 4.5× zoom, as depicted in (**G**). * adapted from Koromila and Stathopoulos, 2019.

**Figure 2 cells-13-01759-f002:**
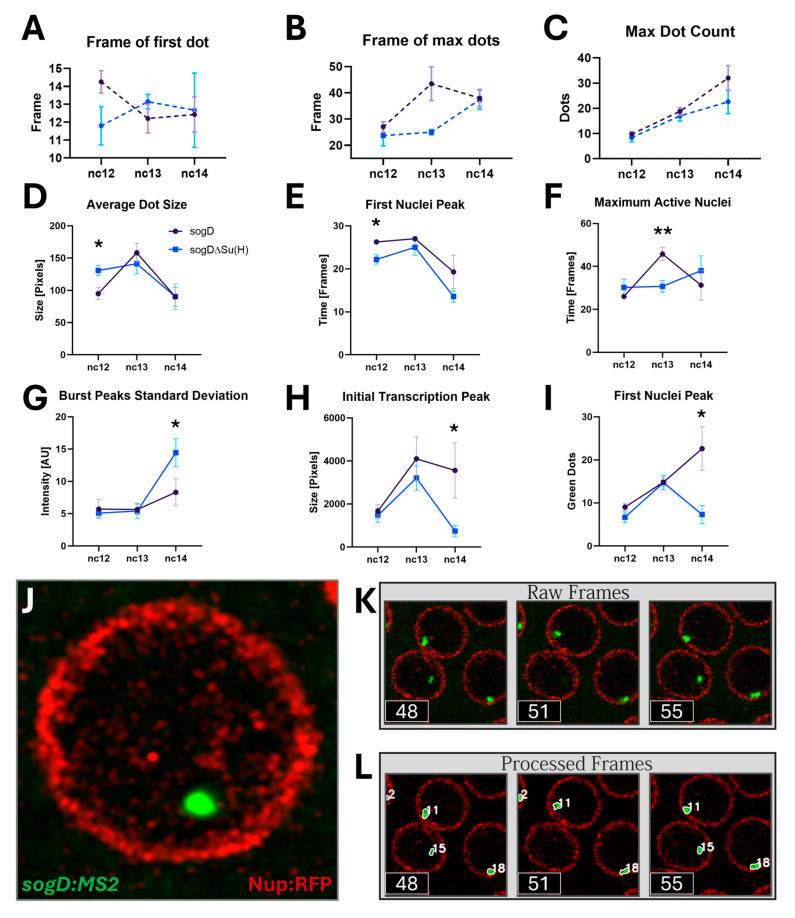
Tissue-level effects of Su(H) on *sogD*-driven expression. (**A**) There is a detectable acceleration in first dot initiation in the absence of Su(H) binding at nc12. (**B**) There is a seeming acceleration in the time to initiation of all dots in the absence of Su(H) at nc13. (**C**) The maximum number of dots in the region measured does not seem to change in response to Su(H) absence. (**D**) Su(H) occupancy at *sogD* reduces average transcription rates during nc12, but not late stages (*: *p* = 0.0295). (**E**) Transcription initiation manifests as a spike in the number of dots; Su(H) occupancy at *sogD* slows progression to this first peak in active nuclei during nc12 (*: *p* = 0.0243). (**F**) Su(H) occupancy at *sogD* decreases the time required to reach the maximum number of active nuclei as the MZT upticks at nc13 (**: *p* = 0.0075). (**G**) Su(H) occupancy at *sogD* reduces the standard deviation of peaks in dot intensity at nc14 (*: *p* = 0.0431). (**H**) Su(H) occupancy at *sogD* increases the total number of dot pixels during transcription initiation during nc14 (*: *p* = 0.0146). (**I**) Su(H) occupancy at *sogD* increases the number of active nuclei during transcription initiation at nc14. (*: *p* = 0.0111) (**J**) Super-resolution imaging facilitates consistent, precise nascent transcript detection. (**K**,**L**) The tracking algorithm is able to follow dots through the frames of each stage, including through bursting events (see dot #15 in frame 51 of (**L**)). (n-values are as follows: *sogD*: nc12 = 4, nc13 = 5, nc14 = 7; *sogD^ΔSu(H^*^)^: nc12 = 5, nc13 = 7, nc14 = 7; error bars are SEM).

**Figure 3 cells-13-01759-f003:**
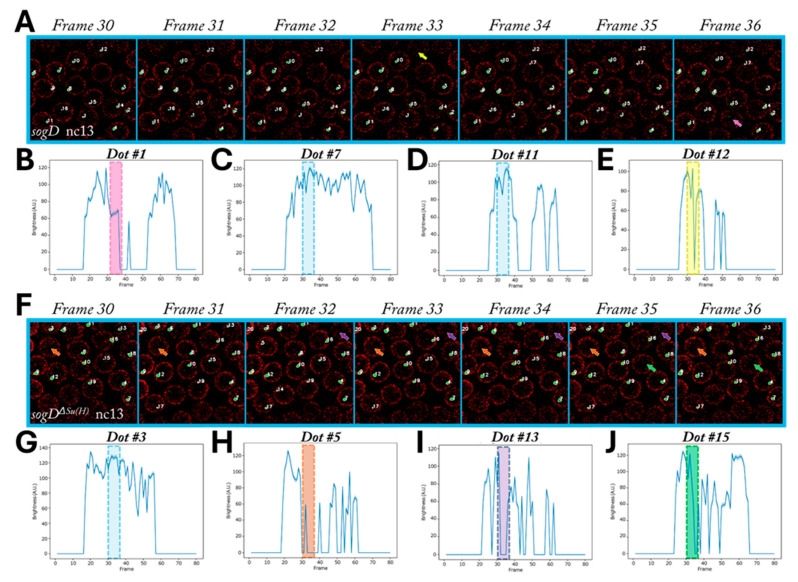
Single-dot tracking reveals stochastic bursting events. (**A**) Seven frames from one wt sogD nc13 movie showing tracking (masks and numeric labels) and bursting events (arrows). (**B**–**E**) Plotted expression dynamics for four dots from the movie in (**A**). Pink (frame 36) and yellow (frame 33) arrows in (**A**) correspond to bursting events within pink and yellow shaded regions in (**B**) and (**E**), respectively. Blue boxes correspond to dots which do not burst within the temporal window shown (**C**,**D**). (**F**) Seven frames from one mutant sogD movie showing tracking (masks and numeric labels) and bursting events (arrows). (**G**–**J**) Plotted expression dynamics for four dots from the movie in (**F**). Orange (frames 30,31,33–36), purple (frames 32–35), and green (frames 35,36) arrows in (**F**) correspond to bursting events within orange, purple, and green shaded regions in (**H**), (**I**), and (**J**), respectively. The blue box in (**G**) corresponds to dot #3, which did not burst within the temporal window shown.

**Figure 4 cells-13-01759-f004:**
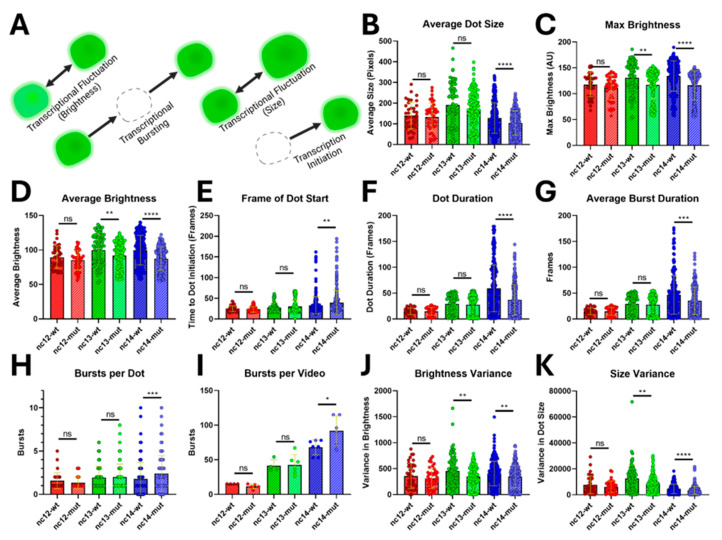
Spatiotemporal fluorescent dot dynamics. (**A**) Fluctuations in dot brightness and size, blinking, and variations in rates of transcription, bursting, and transcriptional initiation, respectively. (**B**) Average dot size is reduced in the mutant at nc14 (**** *p* < 0.0001). (**C**) Maximum dot brightness was reduced in the mutant at nc13 and nc14 (** *p* = 0.001, **** *p* < 0.0001). (**D**) Average dot brightness was reduced in the mutant at nc13 and nc14 (** *p* = 0.037, **** *p* < 0.0001). (**E**) Expression initiates later in the mutant during nc14 (** *p* = 0.0011). (**F**) Dots do not last as long in the mutant during nc14 (**** *p* < 0.0001). (**G**) Transcription between off events is shorter-lived in the mutant during nc14 (*** *p* = 0.0003). (**H**) Dots enter and exit transcription more often in the mutant at nc14 (*** *p* = 0.0003). (**I**) The total number of bursts per movie is increased in the mutant at nc14 (* *p* = 0.0426). (**J**) Dot brightness is more homogeneous in the mutant at nc13 and nc14 (nc13 ** *p* = 0.0047, nc14 ** *p* = 0.0022). (**K**) Dot size is more homogeneous in the mutant at nc13 and nc14 (** *p* = 0.0028, **** *p* < 0.0001). (*p*-values were considered not significant (ns) if they were greater than 0.05).

**Figure 5 cells-13-01759-f005:**
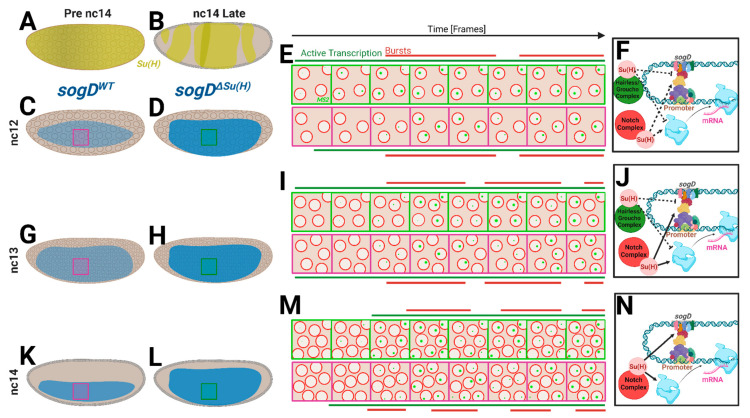
Su(H)/*sogD* regulatory model. (**A**,**B**) Illustrations of Su(H) expression leading up to cellularization (**A**) and during cellularization (**B**). (**C**,**G**,**K**) Illustrations of *sogD*:MS2 reporter expression through stages nc12–14. Magenta boxes correspond to imaged regions and magenta frame illustrations in (**E**,**I**,**M**). (**D**,**H**,**L**) Illustrations of *sogD^ΔSu(H)^*:MS2 reporter expression through stages nc12–14. Green boxes correspond to imaged regions and green frame illustrations in (**E**,**I**,**M**). (**E**,**I**,**M**) Illustrations representing movie progressions of each stage, nc12, 13, and 14, respectively, at the locations of the magenta/green boxes in embryos to the left. Illustrated frames are not to scale and represent broad conceptual representations of observed data. Red bars represent bursting peaks; green bars represent durations where frames have any green dots. (**F**,**J**,**N**) Illustrations of the mechanistic model of action of Su(H) at *sogD*. In the model, Su(H) functions predominantly to promote transcription at nc14, likely through Notch signaling, but has less effect in earlier stages.

## Data Availability

All data included in the manuscript or in the Appendix A are available upon request. Custom code for detecting, tracking, and quantifying dots can be found at https://github.com/jacobluber/hairless_pathway_bursting_to_gastrulation, accessed on 9 October 2024.

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
