# Peer review of "Su(H) Modulates Enhancer Transcriptional Bursting in Prelude to Gastrulation"

_cells, 2024, doi:10.3390/cells13211759_

Round 1

Reviewer 1 Report

Comments and Suggestions for Authors

In this paper, the authors investigate the effect of Suppressor of Hairless [Su(H)] on expression driven by the sog distal enhancer (sogD) in the nc 12-14 Drosophila embryo. They use a combination of the MS2 system and superresolution (Zeiss airyscan) in live embryos to measure the dynamics of transcription of MS2 loops that are driven either by the wildtype sog distal enhancer or by a sogD enhancer with the Su(H) sites mutated. After imaging and analyzing many nuclei from 4-7 embryos each, the authors were able to perform statistical tests on different metrics of the MS2 fluorescence dynamics. They found that Su(H) activity increased from nc 12-14, and it acted primarily to make sog expression stable and robust during nc 14, a fact that is perhaps surprising because Su(H) is viewed as a repressor of sog expression.

This paper is uses advanced live imaging techniques to address a timely question in genetics and developmental biology: how does the enhancer/transcription factor complex interact with the promoter to affect gene expression? As such, their results are important and timely. However, there remain several problems with this manuscript that must be resolved, which would improve the impact of the paper prior to publication.

First, the authors should explain more details about the specifics of the study, not just in the Methods but also in the main text. Specific points:

- Line 176: “mutant conditions”: what are these mutant conditions? From the name of the construct, it looks like the Su(H) sites in the enhancer were mutated. How were they mutated?

- In general: The construct itself was not described. Is it a CRISPR edit to sog itself? A transgene? Using what method of insertion? Is it driving lacZ expression? What is the promoter (e.g., eve min)?

- Line 202: “increases variability between samples.” Why do the authors think that? Is it because the errorbars in Fig 2A are bigger for delta? If so, this should be explicitly stated. A statistical test would be good to make their case. However, a similar test would probably need to be performed on other metrics, because for some graphs in Fig 2, it does not look like the sogD_delta_Su(H) construct has larger errorbars.

- Line 207: What is "deep statistical analysis"?

- in general, there needs to be more explanation of how the metrics in Fig 2 were calculated and why they are connected with the interpretation the authors are giving.

- Line 240: “Interestingly, comparison of our sogD and sogDdeltaSu(H) datasets revealed significant fluctuations in tissue-level transcript density in both reporters at all stages (Sup. Fig. 4A-F).” Why do the authors think so? Because there are errorbars? How did they quantify it? What value (of the quantification) would be expected if there were not “significant” fluctuations? What does “significant” mean?

- Line 242: “This implies that these seemingly random distributions of individual bursting events coalesce to affect tissue-level transcript fluctuations.” How do the authors conclude that?

- In Fig S4, what do the dots represent? Means of all nuclei? And the errorbars? SEM? How do the authors interpret this figure as depicting bursting?

Second, the flow of logic is difficult to follow. Specifically:

- Lines 197-201: The authors talk about “alterations” and “shifts” but don’t explain in which direction. Are they saying removing Su(H) binding sites decreases maturation time or extends it? (This comes up later.)

- It is hard to determine the difference between Fig 2 and 4. Not only do they seem to be the same or similar metrics, only plotted in different ways, but they seem to be telling different stories, which is confusing.

- In the last paragraph of the results section, it would improve readability if the authors gave a bit more context or fleshed out their interpretations more. For example, the sentence starting with “Contrary…” is not explained well. How is it contrary? Are the authors talking about how, without Su(H), sog *would have been* expressed there? Perhaps they could say “dorsal lateral” since sog does not expand really all that far into the dorsal side even when Su(H) is removed. Then they could specifically mention that the “effect” they’re talking about is on sog expression, which would make Su(H) out to be a repressor of sog. But the dots getting smaller make Su(H) seem like an activator. Hence, the “contrary”.

- Going hand-in-hand with the above, the end of the paragraph, starting with the sentence that starts with “Concurrently…” it looks like the authors are trying to say that Su(H) positively regulates sog expression. This becomes clear (is stated explicitly) in the Discussion on Line 276. It would be nice if, in addition to explaining further about what they mean in the “Contrary” sentence, if the authors could more explicitly state here in the results that their finding is exciting because Su(H) looks like it is positively regulating sog. If they did that, it would have some overlap with the Discusison, but I think that’s acceptable because it would provide good context to the reader and help them digest the idea. (I spent a while re-reading this paragraph to make sure I could follow the authors’ logic.)

Third, there are discrepancies between the figures and the text/legends. Specific examples:

- Fig 2C legend says, “The number of dots in the region measured does not seem to change in response to Su(H) absence,” which implies an average or median of number of dots. But Fig 2C is depicted as the Max Dot Count.

- Fig 2D legend says, “Su(H) occupancy at sogD reduces average transcription rates during nc12, but not late stages.” But the metric depicted in the Figure itself is dot size. There needs to be an explicitly stated and well-explained connection between the dot size and transcription rates.

- Fig 2F legend says, “Su(H) occupancy at sogD increases the maximum number of active nuclei as the MZT upticks at nc13.” In this case, the Figure 2F title agrees: “Maximum Active Nuclei”.  However, the y axis label is “Time [Frames].” So is the graph showing when the max number of active nuclei are seen (in which case, Su(H) makes this happen later and the graph does not match the legend), or is the y-axis supposed to be labeled something like “number of nuclei”?

- Figure 2H legend says, “Su(H) occupancy at sogD increases the total number of dot pixels during transcription initiation during nc14 (*: p=.0146).” It is hard to understand how that description connects with “Initial Transcription Peak” (graph title) or “Size [Pixels]” (y-axis label), or how the title and y-label connect with each other. Also, the authors need to describe how they interpret this metric. What does it mean physically? Finally, why does this metric get a p-value, but none of the others do? Is it because this is the only one that isn’t clearly statistically significant (“by eye”), or because it is the only one that really is statistically significant? Or some other reason?

- Figure 2I legend says, “Su(H) occupancy at sogD increases the total number of active nuclei. during transcription initiation at nc14.” It is hard to understand how the legend connects with “First Nuclei Peak” (graph title) or “Green Dots” (y-axis label) , or how the title and y-label connect with each other.

- Line 199: The authors reference Fig 2B to make their point about nc 12, but Fig 2B looks like there is no difference between the two cases at nc 12. Did the authors mean to reference Fig 2A instead?

- Next sentence: I am not sure I agree with the authors’ point here, and part of the reason may be that it's not clear I've interpreted what they're saying correctly. For example, do they see dots earlier with delta Su(H)? A and B both would suggest so, in different ways. But here, they only talk about "alter" and "shift" but they do not say in which way. Furthermore, this sentence says "increased the time" but clearly in both A and B, the delta Su(H) decreases the time. So what are they talking about? The shift from removing Su(H) sites, or from having them?

Several minor points follow.

- Fig 2B doesn't have a mention in the legend, but “C” occurs twice.

- Figure 2G legend should say “at nc 14”.

- Figure 2I: I think there’s an extra period in there.

 - Line 207: Possibly need to add, "measured by dot size"

- Fig 3 legend, 2nd line: why is there a “#” there? Also, which movie?

- Line 242: “Sup. Fig. 4A-F”. In the supplement, there are two Sup Fig. 3’s. So this reference is probably to the second of those, which should be fixed in the supplement.

- Speaking of Sup Fig 4, the axis tick labels should be in bigger font. I had to zoom to 400% to read them.

- Lines 260-261: same kind of genotype, twice

Author Response

Thank you so much for the thorough review; we believe it has significantly improved the manuscript. Please see the attachment.

Reviewer 2 Report

Comments and Suggestions for Authors

sog_Distal (sogD) is an enhancer of the sog gene. It is well-established that Su(H) occupies on sogD enhancer to inhibit the expression of sog. In this study, using the previously established MS2-MCP live imaging system, which can monitor nascent transcript production, and two reporters - sogD:MS2 and sogDΔSu(H):MS2 (Koromila and Stathopoulos, 2019, Cell Reports), the authors examined the effects of Su(H) binding on sogD expression dynamics during stages nc12-14. Through single-nuclei tracking and statistical analyses, the authors found that the transcription of sogD burst more often for shorter durations in the absence of Su(H) binding. Thus, they concluded that Su(H) acts to bolster robust transcription of sog by stabilizing sogD enhancer-promoter contact (EPC), contrary to its well-known role in transcriptional repression dorsally.

Major points:

1. In the abstract, the authors claimed that “Su(H) absence during embryonic development exhibits region-specific effects, inhibiting expression dorsally and enhancing expression ventrally, implying …” (line 25-27). Figure 4 likely supports the conclusion that Su(H) absence inhibits expression dorsally. However, it seems that no evidence was provided to support the conclusion that Su(H) absence enhances expression ventrally.

2. Fig. 3H of the published paper (Koromila and Stathopoulos, 2019) showed that the average dot size is increased in sogDΔSu(H):MS2 reporter from stage nc13 to nc14. However, in Fig. 4B of this study, the quantitative data demonstrated that the average dot size of sogDΔSu(H):MS2 reporter is not changed in stage nc13, and decreased in stage nc14. Is it a discrepancy?

3. It is overstating to make a completely contradictory conclusion that Su(H) functions to stabilize and increase, rather than inhibit, sog expression, solely based on the reporter data and statistical analyses.

Minor points:

1.       Line 260-261: perhaps it is “smaller and less bright in sogDΔSu(H) than in sogD embryos” rather than “smaller and less bright in sogDΔSu(H) than in sogDΔSu(H) embryos”.

Author Response

Thank you and please see the attachment.

Round 2

Reviewer 2 Report

Comments and Suggestions for Authors

the revised manuscript of "Su(H) Modulates Enhancer Transcriptional Bursting in Prelude to Gastrulation" (cells-3237810) is now acceptable for its publication in Cells.